# A Target for Increased Mortality Risk in Critically Ill Patients: The Concept of Perpetuity

**DOI:** 10.3390/jcm10173971

**Published:** 2021-09-02

**Authors:** Jarrod M. Mosier, Julia M. Fisher, Cameron D. Hypes, Edward J. Bedrick, Elizabeth Salvagio Campbell, Karen Lutrick, Charles B. Cairns

**Affiliations:** 1Department of Emergency Medicine, College of Medicine, University of Arizona, Tucson, AZ 85724, USA; chypes@arizona.edu; 2Division of Pulmonary, Allergy, Critical Care and Sleep, Department of Medicine, College of Medicine, University of Arizona, Tucson, AZ 85724, USA; 3Statistics Consulting Laboratory, BIO5 Institute, University of Arizona, Tucson, AZ 85721, USA; julia@statlab.bio5.org; 4Center for Biomedical Informatics and Biostatistics, University of Arizona Health Sciences, Tucson, AZ 85721, USA; edwardjbedrick@arizona.edu; 5College of Medicine, University of Arizona, P.O. Box 245017, 1501 N. Campbell Ave, Tucson, AZ 85724, USA; bsalvag@arizona.edu (E.S.C.); klutrick@arizona.edu (K.L.); 6Department of Family and Community Medicine, College of Medicine, University of Arizona, Tucson, AZ 85711, USA; 7College of Medicine, Drexel University, 2900 W Queen Lane, Philadelphia, PA 19129, USA; cbc77@drexel.edu

**Keywords:** perpetuity, acuity, emergency department, critical care, critically ill, mechanical ventilation, intubation

## Abstract

Background: Emergency medicine is acuity-based and focuses on time-sensitive treatments for life-threatening diseases. Prolonged time in the emergency department, however, is associated with higher mortality in critically ill patients. Thus, we explored management after an acuity-based intervention, which we call perpetuity, as a potential mechanism for increased risk. To explore this concept, we evaluated the impact of each hour above a lung-protective tidal volume on risk of mortality. Methods: This cohort analysis includes all critically ill, non-trauma, adult patients admitted to two academic EDs between 1 November 2013 and 30 April 2017. Cox models with time-varying covariates were developed with time in perpetuity as a time-varying covariate, defined as hours above 8 mL/kg ideal body weight, adjusted for covariates. The primary outcome was the time to in-hospital death. Results: Our analysis included 2025 patients, 321 (16%) of whom had at least 1 h of perpetuity time. A partial likelihood-ratio test comparing models with and without hours in perpetuity was statistically significant (χ^2^(3) = 13.83, *p* = 0.0031). There was an interaction between age and perpetuity (Relative risk (RR) 0.9995; 95% Confidence interval (CI_95_): 0.9991–0.9998). For example, for each hour above 8 mL/kg ideal body weight, a 20-year-old with 90% oxygen saturation has a relative risk of death of 1.02, but a 40-year-old with 90% oxygen saturation has a relative risk of 1.01. Conclusions: Perpetuity, illustrated through the lens of mechanical ventilation, may represent a target for improving outcomes in critically ill patients, starting in the emergency department. Research is needed to evaluate the types of patients and interventions in which perpetuity plays a role.

## 1. Introduction

Emergency department (ED) visits in the United States totaled over 146 million in 2016, a 12% increase over five years [1,2]. The percentage of those patients requiring critical care, time providing critical care, and admissions to the intensive care unit (ICU) has increased disproportionately, despite decreasing ED- and ICU-bed availability [3,4,5]. This prolongs ED boarding and creates difficult ICU triage decisions that can lead to delayed or inappropriate care and poorer outcomes [6,7,8,9,10,11,12]. A recent study reported improved outcomes with a dedicated ICU-level area in the ED [13], suggesting a need to identify contributing factors and targets for interventions to reduce this risk.

Conceptually, the flow of time-related events in ED care is one potential factor. Patients are treated based on triage acuity and time-sensitive interventions for high-acuity processes (e.g., sepsis, stroke, myocardial infarction, and cardiac arrest) are performed to attenuate, or even reverse, the propagation of disease. There is not, however, enough dependable data on how outcomes are shaped by the care that follows such interventions. Data are increasingly important as the transition between the ED and ICU becomes blurred.

Acute respiratory failure represents one such paradox. Most patients with acute respiratory failure present to the ED. In many of these patients, early intubation is associated with improved outcomes [14], and is performed earlier with higher acuity. Mechanical ventilation after intubation, however, is often suboptimal [15,16,17,18], persists into the ICU, and is closely associated with outcomes [18,19,20,21].

We conceptualized care provided after an acuity-based intervention as the “*perpetuity*” of disease, where suboptimal or even injurious management perpetuates risk from critical illness, and hypothesized that it may play a role in mortality. We aimed to explore the concept of perpetuity by evaluating one type of management known to be associated with affecting outcomes after a defined acuity-based intervention: ventilator management after intubation.

## 2. Methods

### 2.1. Study Setting

This study was conducted at two academic medical centers where the EDs are staffed with emergency medicine faculty and residents, and the ICUs are staffed with pulmonary and critical care faculty, fellows, and internal medicine residents. The Banner–University Medical Center Tucson (BUMCT) is a Level 1 trauma center with an annual ED census of 85,000. The Banner–University Medical Center South (BUMCS) has an annual ED census of 54,000. Neither hospital operates a formal ED–ICU. After the decision is made to transition care from the ED to the ICU, the ICU team is responsible for the patient’s care, including the remaining time that the patient is boarded in the ED.

All ED patients requiring ICU care are incorporated into a quality improvement registry, which is maintained by data downloaded from the hospitals’ electronic medical record (EPIC systems, Verona, WI, USA). This project adhered to the STROBE reporting guidelines and Patient-Centered Outcomes Research Institute (PCORI) standards for registry studies [22], and it was granted exemption from informed consent and approved by the University of Arizona Institutional Review Board (#1607695679).

### 2.2. Study Design

This is a retrospective cohort analysis of adult patients (>18 years), intubated in the prehospital setting or ED, and admitted to the ICU for non-traumatic critical illness at two academic hospitals between 1 November 2013 and 30 April 2017 and followed until death or hospital exit (whichever came first). We excluded patients intubated in the ICU (see Appendix A: Methods). We defined a period of time as contributing to perpetuity if the patient was charted as receiving tidal volumes > 8 mL/kg predicted body weight over that period of time after intubation. Tidal volumes are documented at the initiation of mechanical ventilation and then semi-regularly (with an hourly goal) by the respiratory therapy staff. We followed the patient until death or hospital exit (whichever came first).

### 2.3. Calculation of Time in Perpetuity

The exact amount of time each patient spent in perpetuity cannot be known because we do not have continuous tidal volume readings. We thus approximated time in perpetuity as follows: at every time stamp, we determined whether or not the tidal volume was >8 mL/kg. If it was, we added the period of time between the previous and current time stamps to the previous total time in perpetuity. The new total time in perpetuity was designated as the value of perpetuity at that time stamp. Thus, time in perpetuity either remained the same or increased at every time stamp, depending on whether the observed tidal volume at that time stamp was ≤8 mL/kg or >8 mL/kg, respectively. The accumulation of perpetuity began at intubation or ED arrival, whichever came last. It stopped accumulating at ICU exit, death, or hypothesized extubation, whichever came first (Figure 1). The primary outcome was time to in-hospital death.

Figure 1 Legend: This figure demonstrates how perpetuity is calculated for a hypothetical patient given the preprocessing assumptions. The upper panel shows the tidal volumes (y-axis) and the lower panel shows the hours accumulated in perpetuity (y-axis) over time (x-axis). Time zero is intubation time, and intervals between charted tidal volumes (solid dots) accumulate perpetuity time if they are >8 mL/kg. For this patient, each tidal volume charted for the first 6 h was >8 mL/kg and accumulated time in perpetuity. However, no time was accumulated after that until 18–25 h, when tidal volumes again crossed the threshold. The large gap between 25 h and the next charted tidal volume exceeds the cutoff threshold for assumed extubation and reintubation and thus does not contribute to perpetuity time.

### 2.4. Hypothesized Extubation

To mitigate potential issues with inflating time in perpetuity, we made assumptions regarding long durations between successively charted tidal volumes. Specifically, we were concerned that long intervals could have resulted from undocumented extubations. After an analysis of time intervals, and accounting for typical staff shift hours, we chose two durations (eight and twelve hours) between charted tidal volumes as cut-offs (see Appendix A). No periods of time over the 8 mL/kg threshold after the first interval of greater than eight or twelve hours were included in the perpetuity calculation. We did this for two primary reasons: (a) we did not want to include tidal volumes possibly resulting from re-intubations, and (b) we did not want to include long periods of unintubated time in the total time in perpetuity (Figure 1). Each cut-off choice resulted in the creation of a different data set.

### 2.5. Sensitivity Analysis

We conducted a sensitivity analysis to better explore the effects of our preprocessing decisions, specifically those related to hypothesized extubation and the calculation of time in perpetuity. These preprocessing choices could affect later inference; thus we examined the data sets for the two choices. We explored the effect of the choices of eight and twelve hours as cut-offs by allowing the cut-offs to be four, five, six, seven, and twenty-four hours. We explored the effect of disallowing further incrementation of hours in perpetuity after the first cut-off interval by (a) allowing intervals of all sizes at any point in time to increment the time in perpetuity (no adjustment data set) and (b) allowing only intervals less than six hours long to increment the time in perpetuity (gaps ≥ 6 h removed data set). The key difference between the “gaps ≥ 6 h removed” data set and the six-hour cut-off dataset is that in the first, but not the second, small intervals after the first long interval are still allowed to increment the time in perpetuity. We examined the effect of assuming a constant charted tidal volume in large intervals by instead interpolating linearly between the first and second tidal volumes in intervals greater than or equal to six hours (linear interpolation in gaps ≥ 6 h data set). Only the portion of the intervals predicted to be over the 8 mL/kg threshold was added to the time in perpetuity. We explored the effect of propagating charted tidal volumes backward by instead assuming the last charted tidal volume carried forward (last-one-forward data sets), see Appendix A.

### 2.6. Data Analysis

Cox models with time-varying covariates were fit to the eight- and twelve-hour cut-off data. The models had identical specifications. The outcome variable was time to death during the hospital stay. Patients were censored at hospital discharge. The primary predictor of interest was hours in perpetuity, which was treated as a time-varying covariate. Other predictors that are potential confounders or indicate severity of illness were chosen, and included age, initial oxygen saturation at triage, hours in acuity, mode of arrival (ground transport, air transport, private vehicle, and other), hospital campus (BUMCT or BUMCS), whether or not the patient was on non-invasive positive pressure ventilation prior to intubation, ideal body weight, body mass index, and the Emergency Severity Index score at ED triage. Heart rate, systolic blood pressure, and mean arterial pressure were included as static covariates set to their first recorded values. Two interactions we thought were clinically important were included a priori as well: the interaction of hours in perpetuity with age and the interaction of hours in perpetuity with oxygen saturation.

Penalized splines were used to handle nonlinearity of continuous predictors [23]. To decide which predictors were best included as nonlinear, we fit a separate model per continuous predictor. In each of these models, we included a single predictor as nonlinear and allowed the optimal degrees of freedom for the nonlinear portion to be estimated with Akaike’s Information Criterion [24]. All other predictors in these models were left as linear. We tested whether that nonlinear portion of the spline predictor was statistically significant, at a 5% false positive rate, using the Wald χ^2^ test. We next fit a model that included, as nonlinear, all predictors indicated by the previous step. For each spline term, we set the degrees of freedom to be the previously identified optimal degrees of freedom. The interactions of hours in perpetuity with age and oxygen saturation were included in this model as well. As a final check on the form of the nonlinear predictors, we again tested the statistical significance of the nonlinear portion of each spline term. Only those spline terms with nonlinear portions statistically significant at a 5% false positive rate (hours in acuity and heart rate) were included as nonlinear in the final model.

An additional (reduced) model was fit to each data set that did not include any of the terms involving time in perpetuity (the main effect of hours in perpetuity and both interactions with hours in perpetuity). The full and reduced models for a given data set were compared using a partial likelihood-ratio test in order to test whether the inclusion of perpetuity in its three forms resulted in a meaningfully better model fit. For the sensitivity analysis, only the final full and reduced models described above were fit to each data set.

Data processing and statistical analyses were conducted in the R statistical computing language [25]. Data manipulation was handled with the tidyr and dplyr packages [26,27], plots were created with the ggplot2 package [28], and results tables were constructed with the xtable package [29]. The Cox models with time-varying covariates were fit with the coxph function in the survival package [30,31].

We conducted a simple power calculation in order to gain insight into the number of participants needed to detect various hazard ratios corresponding to a one-hour increase in perpetuity time. Our approach follows van Belle in assuming survival times are exponentially distributed [32]. We assume we have two groups of participants, one with one hour longer in perpetuity than the other. Then, with 1000 participants per group (2000 participants in total) and a 5% significance level, we will have ≥80% power to detect a hazard ratio less than 0.89 or greater than 1.13. As (a) hazard ratios of 0.89 and 1.13 are fairly small, and (b) we collected data on around 2000 participants, we are confident that the study is well-powered.

## 3. Results

Our final analyses were based on 2025 participants (Figure 2). Demographic data are found in Table 1. Sixteen percent (321) patients had at least one hour of perpetuity time. For both the eight- and twelve-hour cut-off models, the following were statistically significant (all *p* < 0.05): the interactions of age/perpetuity and oxygen saturation/perpetuity as well as the main effects of perpetuity, age, oxygen saturation, hours in acuity (the nonlinear component), campus, non-invasive positive pressure ventilation, heart rate (the nonlinear component), and triage score, see Table 2 and Table 3 for estimated coefficients. Note that systolic blood pressure and mean arterial pressure were highly correlated ($\hat{\rho }$ = 0.92), and in models that left out either predictor, the other was statistically significant (*p* < 0.05). For both models, a partial likelihood-ratio test indicated that the inclusion of perpetuity significantly improved model fit (eight-hour model: χ^2^ = 13.83, df = 3, *p* = 0.0031; twelve-hour model: χ^2^ = 8.61, df = 3, *p* = 0.0348—Appendix B Table A2 and Table A4).

Because perpetuity occurs in the models in two interactions and as a main effect, it is easiest to see its impact by fixing age and oxygen saturation and then examining the hazard ratio associated with a one-hour increase in perpetuity for those fixed values.

Our results show that for a fixed age, the hazard of dying in the hospital decreases with an increase in oxygen saturation and decreases with increasing age for a fixed oxygen saturation. In examining the effect of perpetuity, these patterns indicate that for younger people and for older people with low/moderate oxygen saturation, a one-hour increase in perpetuity is associated with an increased hazard of dying in the hospital. This pattern holds across both primary models, although it is stronger for the eight-hour model (Figure 3 and Figure 4).

The sensitivity analyses showed that the inclusion of hours in perpetuity resulted in a statistically significant improvement in the model fit for the four-, six-, and seven-hour cut-off models and in the eight-hour last-one-forward model (Appendix B). However, perpetuity did not significantly improve the model fit for the five- and twenty-four-hour cut-off models, the no-adjustment model, the model with intervals ≥ six hours removed from perpetuity, the model with linear interpolation in intervals ≥ six hours, and the twelve-hour last-one-forward model.

## 4. Discussion

The aim of this study was to explore the concept of perpetuity using an aspect of mechanical ventilation widely considered to be injurious (high tidal volumes) that can only occur after an acuity-based intervention (intubation). Our results support the concept of perpetuity as a contributing factor to risk after acuity-based interventions, where risk accumulates from inappropriate, inadequate, or injurious management after that intervention and is indefinite unless something changes. To our knowledge, only two other studies have shown an increase in risk of death per unit of time receiving injurious mechanical ventilation; one with tidal volumes >8 mL/kg for >24 h [33], and the other with injurious driving pressure or mechanical power per day [34]. Our results extend this knowledge to include accumulated risk per hour of injurious mechanical ventilation, even when discontinuous.

Interestingly, we found that younger age and more severe hypoxemia poses more risk from time in perpetuity. We hypothesize that those patients may have respiratory failure etiologies (e.g., acute respiratory distress syndrome [ARDS]) that tend to reduce lung compliance and thus increase susceptibility to perpetuity-based injury (i.e., ventilator-induced lung injury [VILI]).

Given the concerning data on mechanical ventilation in the ED as well as the debate about its management, we used intubation as our acuity-based intervention and mechanical ventilation as our perpetuity-based management [35,36]. Patients require intubation for many etiologies, and the higher the risk of respiratory arrest, the sooner the patient will be intubated (i.e., acuity). However, mechanical ventilation in the ED has proved to be troublesome. Lung protective ventilation is used infrequently and some patients progress to ARDS shortly after hospital admission, indicating VILI may contribute [15,16,17]. A recent study showed that most patients do not receive lung protective ventilation while in the ED and are less likely to have ventilator adjustments during times of ED strain [18]. Inertia dictates that management strategies in the ED often carry over into the ICU for significant periods of time [18,19,20].

High tidal volume is an attractive option for evaluating perpetuity as volutrauma is known to be injurious regardless of the precipitating requirement for intubation [37,38,39]. While tidal volumes may only be injurious in proportion to reductions in lung compliance, and mechanical power research is evolving our understanding of ventilator-induced lung injury [40,41,42,43], large tidal volumes are still widely considered to be injurious. Unfortunately, PEEP values were not available in our dataset. Given the observed interaction with age and degree of hypoxemia, it is possible that our association is an underestimation of the risk of perpetuity. Future research should evaluate this concept using driving pressure and mechanical power, adjusted for severity of lung injury.

Another example of perpetuity is antibiotic timing in sepsis. The data on timing of antibiotics have some limitations, but generally indicated earlier appropriate antibiotics improves outcomes. Thus, initiatives mainly focus on initial antibiotic timing. However, the second dose is delayed in one third of patients, which worsens outcomes irrespective of the timing of the first dose and, paradoxically, more commonly with optimal first-dose timing [44]. Studies are needed to explore what other interventions and in what other diseases, perpetuity may play a role in accumulated risk over time.

Our sensitivity analyses showed that the inclusion of hours in perpetuity resulted in a statistically significant improvement in the model fit for the four-hour, six-hour, and seven-hour cut-off models and the last-one-forward eight-hour cut-off model, but not for the five- and twenty-four-hour and no-adjustment models, the model where intervals ≥ six hours were removed from perpetuity, the model with linear interpolation in intervals ≥ six hours, and the last-one-forward 12 h cut-off model. The first conclusion that can be reached is that while the results from the eight- and twelve-hour models are not due to finely tuned preprocessing choices; preprocessing does have the ability to affect inference. Second, the no-adjustment model results are not entirely unexpected. Assuming that long intervals between successive tidal volumes are indicative of extubation and possible reintubation, the no-adjustment data set leaves possibly large periods of extubation in some perpetuity calculations. This could result in substantial noise in the data.

The five- and twenty-four-hour cut-off results initially appear to be outliers given that all the other basic cut-off models resulted in the same inference. However, the χ^2^ values for the five-, six-, twelve-, and twenty-four-hour cut-off models are all close to the test’s critical value. Thus, while the inferences for these tests differ, the results are not substantially different.

The results for the remaining data sets are more challenging to explain. However, they highlight the need for documentation of (1) extubation and reintubation, (2) changes to the ordered tidal volume, and (3) the use of spontaneous breathing mode. Having these three types of information could dramatically reduce the noise in the data by giving more information on the patients’ charted tidal volumes at each point in time. This could lead to more accurate estimates of the time in perpetuity and its effect on the risk of dying in the hospital.

The time to ICU-level of care, not the time to the ICU, improves outcomes, and delaying perpetuity-focused care until ICU admission is a missed opportunity [13,45]. Gunnerson and colleagues found that an ED-based ICU program reduced 30-day mortality and ICU admissions [13]. These findings, along with our results, and those by Sjoding [33], and Leisman [44], suggest that targeted interventions focused on reducing perpetuity are opportunities to improve outcomes for patients regardless of their physical location.

Clinical and philosophical implications of perpetuity are potentially immense. As the critical care requirements of ED patients increases, there is substantial debate about what the response to this burden should entail. EDs provide an increasing majority of hospital associated medical care [46], and are under pressure to transfer care to admitting services after completing time-sensitive goals. The American College of Emergency Physicians (ACEP) has a policy statement that “the ED should not be utilized as an extension of the ICU and other inpatient units for admitted patients because this practice adversely affects patient safety, quality, and access to care” [47]. Similarly, the American Academy of Emergency Medicine (AAEM) has a policy statement that critically ill patients should be transferred to the ICU within six hours of arrival to the ED, as “further delay can deplete the ED of resources” [48]. While ACEP believes that hospitals have the responsibility to provide the appropriate inpatient beds and staffing [47], critically ill patients remain at high risk of delayed critical care delivery, and our data suggest perpetuity-based injury contributes to poor outcomes. More than one third of ICU patients spend more than six hours in the ED, which is not based on physiologic or outcome parameters, rather it is the mean boarding time for critical care beds in overcrowded hospitals [9]. Some intubated patients may have shorter stays in the ED [49], but time to ICU admission is associated with mortality even when boarding time is significantly fewer than six hours [50].

Our study has several limitations. We evaluated the effect of duration in perpetuity (time receiving tidal volumes > 8 mL/kg) on mortality, rather than the effect of the intervention itself (incidence of tidal volumes > 8 mL/kg or the magnitude of tidal volumes) or the magnitude of excess tidal volumes. Given the potential influence of the magnitude of tidal volumes in addition to time, we fit a model where the time in perpetuity is weighted by the tidal volume’s percent over the 8 mL/kg threshold (Appendix B, Table A3 and Table A5). Results for these models parallel results for the last-one-forward eight- and twelve-hour models on which they are based, with relative risk of weighted time in perpetuity of 3.7723 (1.2652, 11.2468) in the 8 h cutoff model.

Our results could be influenced by the intervention itself, and most importantly, by the limitations of time-stamped data and the assumptions made in preprocessing. Each assumption was made in the context of the most likely clinical explanation and biased against the hypothesis. We performed several sensitivity analyses to ensure that our assumptions did not inadvertently affect the results. Sensitivity results indicated that pre-processing does affect inference but that many preprocessing choices lead to the conclusion that time in perpetuity impacts the risk of in-hospital mortality. Regardless, even digitally time-stamped data should be interpreted with caution, and the sensitivity analyses may not have adequately detected concerns with the assumptions.

Furthermore, time in perpetuity was accumulated at any point during the initial mechanical ventilation period and it is unclear if early perpetuity is more, or less, injurious than perpetuity that occurs later in the course. Missing data presents another limitation. Eleven percent of patients (242/2267) meeting our inclusion criteria were missing data. Of those, 176 patients died shortly after arrival in the ED (median time 24 min); and no charted tidal volumes were available on those patients. These patients could have biased our results as they were not included in our analysis.

In summary, our results from this large dataset of critically ill patients suggest the presence of a time-based risk from high tidal volume ventilation that varies based on age and degree of hypoxemia. This time-dependent risk, which we term perpetuity, is a potential target in emergency and critical care research design as well as clinical care to improve outcomes. 

This work was presented at the National Foundation of Emergency Medicine annual meeting at the SAEM meeting in Las Vegas, Nevada, in May 2019.

## Figures and Tables

**Figure 1 jcm-10-03971-f001:**
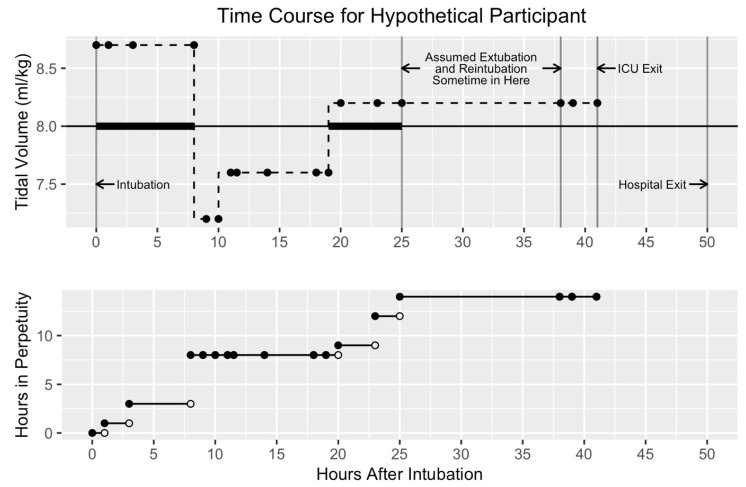
Time course for a hypothetical patient.

**Figure 2 jcm-10-03971-f002:**
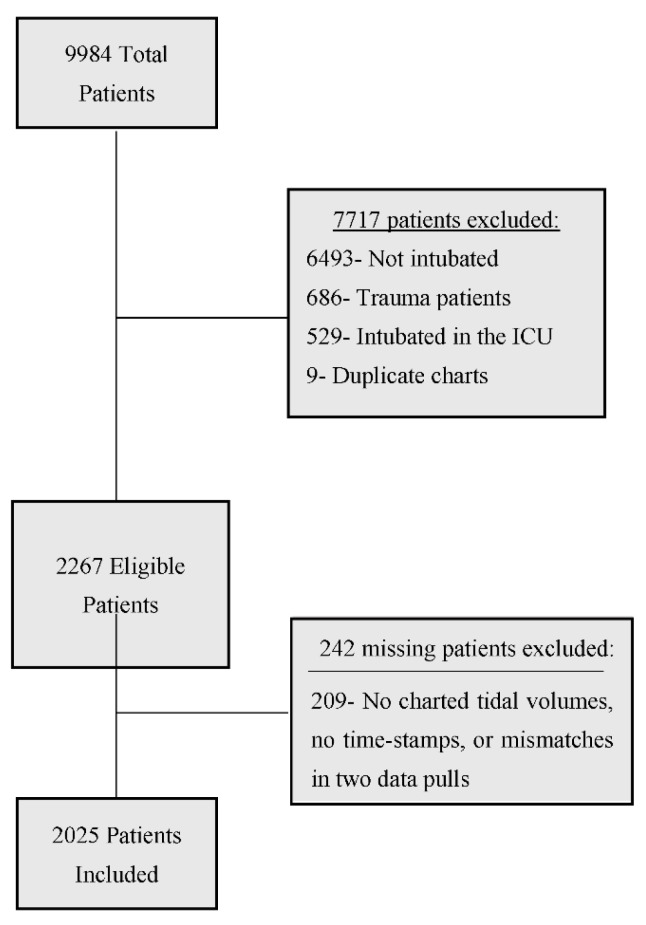
Patient flow chart. Other includes: missing or clearly incorrect height, weight, systolic blood pressure, mean arterial blood pressure, questionable death status, having a documented hospital discharge before ICU discharge, and charted tidal volumes that occurred after ICU discharge.

**Figure 3 jcm-10-03971-f003:**
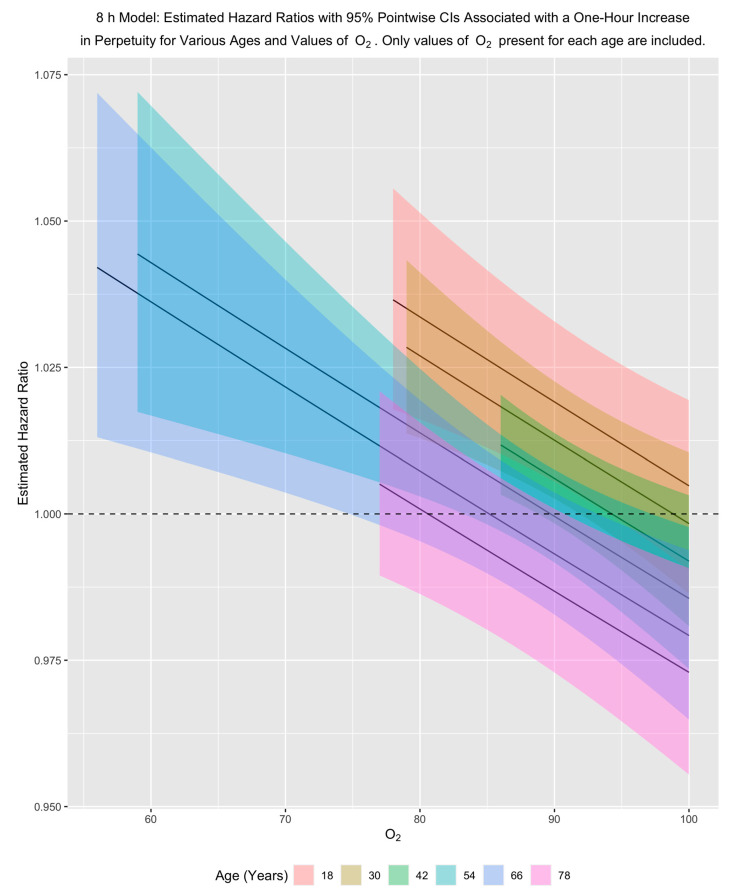
8 h model: Estimated adjusted hazard ratios with 95% pointwise confidence intervals associated with a one-hour increase in perpetuity for various ages and values of O_2_. Only values of O_2_ present for each age are included in the plots. O_2_ is triage oxygen saturation.

**Figure 4 jcm-10-03971-f004:**
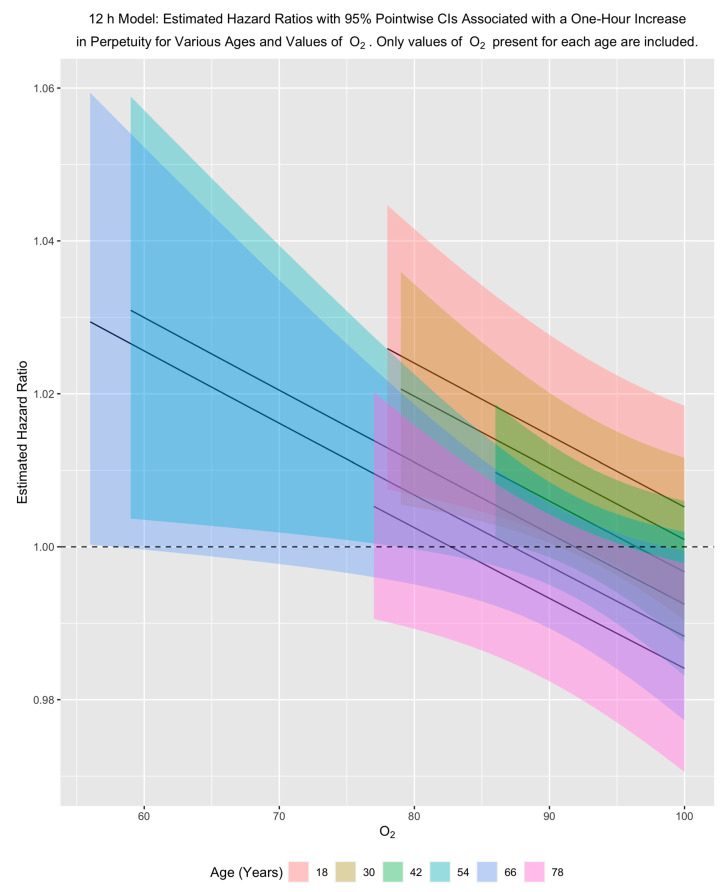
12 h model: estimated adjusted hazard ratios with 95% pointwise confidence intervals associated with a one-hour increase in perpetuity for various ages and values of O_2_. Only values of O_2_ present for each age are included in the plots. O_2_ is triage oxygen saturation.

**Table 1 jcm-10-03971-t001:** Demographics.

Characteristic	Number (Total *n* = 2025)	Percent or Mean (SD)
Demographics		
Age, mean (SD)	--	56 (18)
Male	1198	59%
Height (cm)	--	170.5 (10.2)
Weight (kg)	--	82.2 (26.8)
Ideal Body Weight	--	64.6 (10.6)
Body Mass Index	--	28.3 (9.2)
Mode of arrival		
EMS Ground	1621	80%
Private Vehicle	251	12%
EMS Air	83	4%
Other	70	3%
Campus		
BUMCT *	1282	63%
BUMCS *	743	37%
Hospital Shift at ED Arrival		
Day Shift (7 a.m.–7 p.m.)	1146	57%
Night Shift (7 p.m.–7 a.m.)	879	43%
Admitting Diagnosis		
Neurologic/psychiatric	557	28%
Respiratory, all causes	525	26%
Cardiac arrest	204	10%
Infectious/Allergic/Immunologic	203	8%
Toxin/Toxidrome	156	7%
Cardiovascular	134	6%
GI	112	2%
Renal/GU	39	2%
Endocrine	33	1%
Hematologic/Oncologic	29	1%
Musculoskeletal	15	1%
Trauma	13	1%
No Diagnosis Given	5	0%
Location of Intubation		
Prehospital	269	13%
Emergency Department	1756	87%
Noninvasive ventilation use	315	16%
Emergency Severity Index		
1	940	46%
2	985	49%
3	98	5%
4	2	<1%
Vital Signs		
Heart Rate	--	105 (28.5)
Systolic Blood Pressure	--	133.2 (36.4)
Diastolic Blood Pressure	--	98.4 (24.7)
Mean Arterial pressure	--	96.5 (27.1)
Oxygen Saturation	--	93.3 (10.5)
Acuity ^#^ Minutes	--	99.9 (167.7)
Acuity Hours	--	1.7 (2.8)
Outcomes		
Discharged Alive	1570	78%
In-hospital death	455	22%

Standard deviation [SD]; Emergency medical services [EMS]; * BUMCT (Banner–University Medical Center Tucson); BUMCS (Banner University Medical Center South); Emergency Department [ED]; Gastrointestinal [GI]; Genitourinary [GU]. ^#^ Acuity is the time between ED arrival and the time of intubation.

**Table 2 jcm-10-03971-t002:** 8 h Cut-Off Model: Cox model with hours in perpetuity as a time-varying covariate.

	Estimated Coefficient (95% CI)	Relative Risk (95% CI)	χ^2^	df	*p* Value
Hours in Perpetuity (>8 mL/kg)	0.1558 (0.0737, 0.2379)	1.1686 (1.0765, 1.2686)	13.8442	1	0.0002
Age	0.028 (0.0219, 0.0342)	1.0284 (1.0221, 1.0348)	79.0050	1	0.0000
O_2_	−0.0135 (−0.02, −0.0071)	0.9865 (0.9802, 0.993)	16.7289	1	0.0000
Hours in Acuity (Linear)	0.0148 (−0.042, 0.0717)	1.015 (0.9589, 1.0743)	0.2621	1	0.6087
Hours in Acuity (Nonlinear)			9.3494	3	0.0250
Arrival by EMS Ground	0.0645 (−0.4077, 0.5367)	1.0666 (0.6652, 1.7104)	0.0717	1	0.7889
Other Mode of Arrival	0.0459 (−0.6304, 0.7222)	1.047 (0.5324, 2.059)	0.0177	1	0.8942
POV Mode of Arrival	−0.2503 (−0.8365, 0.3359)	0.7786 (0.4332, 1.3992)	0.7004	1	0.4026
BUMCS	0.4478 (0.248, 0.6476)	1.5648 (1.2814, 1.9109)	19.2972	1	0.0000
NIPPV prior to intubation	−0.9528 (−1.2801, −0.6255)	0.3857 (0.278, 0.535)	32.5512	1	0.0000
Ideal Body Weight	−0.0016 (−0.0111, 0.0078)	0.9984 (0.9889, 1.0079)	0.1161	1	0.7333
BMI	0.0015 (−0.0087, 0.0117)	1.0015 (0.9913, 1.0117)	0.0788	1	0.7789
Heart Rate (Linear)	7 × 10^−4^ (−0.0022, 0.0036)	1.0007 (0.9978, 1.0036)	0.2327	1	0.6296
Heart Rate (Nonlinear)			11.6439	2	0.0033
Systolic Blood Pressure	−0.0024 (−0.0089, 0.0042)	0.9976 (0.9911, 1.0042)	0.4916	1	0.4832
Mean Arterial Pressure	−0.0052 (−0.014, 0.0037)	0.9949 (0.9861, 1.0037)	1.2898	1	0.2561
Acuity Score	−0.4362 (−0.6355, −0.2368)	0.6465 (0.5297, 0.7891)	18.3887	1	0.0000
Hours in Perpetuity × Age	−5 × 10^−4^ (−9 × 10^−4^, −2 × 10^−4^)	0.9995 (0.9991, 0.9998)	7.4002	1	0.0065
Hours in Perpetuity × O_2_	−0.0014 (−0.0023, −6 × 10^−4^)	0.9986 (0.9977, 0.9994)	10.8938	1	0.0010

Confidence interval [CI]; Noninvasive positive-pressure ventilation [NIPPV]; Privately-owned vehicle [POV]; Body mass index [BMI]. The estimated coefficient is on the log(hazard rate) scale. A relative risk over one indicates that having the associated covariate (or each unit of the associated covariate) increases the hazard of dying in the hospital. The nonlinear portion of the spline terms for hours in acuity and heart rate do not have single estimated coefficients. Thus, only χ^2^ tests and the associated *p* values are given for these rows. 2025 patients are included in this model.

**Table 3 jcm-10-03971-t003:** 12 h Cut-Off Model: Cox model with hours in perpetuity as a time-varying covariate.

	Estimated Coefficient (95% CI)	Relative Risk (95% CI)	χ^2^	df	*p* Value
Hours in Perpetuity (>8 mL/kg)	0.1042 (0.0254, 0.183)	1.1099 (1.0257, 1.2009)	6.7183	1	0.0095
Age	0.0279 (0.0217, 0.0341)	1.0283 (1.022, 1.0347)	78.1274	1	0.0000
O_2_	−0.014 (−0.0205, −0.0076)	0.9861 (0.9797, 0.9924)	18.2653	1	0.0000
Hours in Acuity (Linear)	0.0145 (−0.0423, 0.0712)	1.0146 (0.9586, 1.0738)	0.2491	1	0.6177
Hours in Acuity (Nonlinear)			9.4794	3	0.0236
Arrival by EMS Ground	0.0743 (−0.3989, 0.5475)	1.0771 (0.671, 1.729)	0.0947	1	0.7583
Other Mode of Arrival	0.0663 (−0.6111, 0.7437)	1.0685 (0.5427, 2.1037)	0.0368	1	0.8479
POV Mode of Arrival	−0.2338 (−0.8186, 0.351)	0.7915 (0.4411, 1.4205)	0.6140	1	0.4333
BUMCS	0.4415 (0.242, 0.6409)	1.555 (1.2738, 1.8983)	18.8148	1	0.0000
NIPPV prior to intubation	−0.94 (−1.2658, −0.6142)	0.3906 (0.282, 0.5411)	31.9737	1	0.0000
Ideal Body Weight	−6 × 10^−4^ (−0.01, 0.0089)	0.9994 (0.99, 1.0089)	0.0141	1	0.9054
BMI	0.0011 (−0.009, 0.0112)	1.0011 (0.991, 1.0113)	0.0442	1	0.8334
Heart Rate (Linear)	7 × 10^−4^ (−0.0021, 0.0036)	1.0007 (0.9979, 1.0036)	0.2398	1	0.6244
Heart Rate (Nonlinear)			11.4993	2	0.0035
Systolic Blood Pressure	−0.0025 (−0.0091, 0.0041)	0.9975 (0.991, 1.0041)	0.5474	1	0.4594
Mean Arterial Pressure	−0.0049 (−0.0137, 0.004)	0.9951 (0.9863, 1.004)	1.1622	1	0.2810
Acuity Score	−0.4323 (−0.6313, −0.2332)	0.649 (0.5319, 0.792)	18.1107	1	0.0000
Hours in Perpetuity × Age	−4 × 10^−4^ (−7 × 10^−4^, 0)	0.9996 (0.9993, 1)	4.4328	1	0.0353
Hours in Perpetuity × O_2_	−9 × 10^−4^ (−0.0017, −1 × 10^−4^)	0.9991 (0.9983, 0.9999)	5.1694	1	0.0230

The estimated coefficient is on the log(hazard rate) scale. A relative risk over one indicates that having the associated covariate (or each unit of the associated covariate) increases the hazard of dying in the hospital. The nonlinear portion of the spline terms for hours in acuity and heart rate do not have single estimated coefficients. Thus, only χ^2^ tests and the associated *p* values are given for these rows. 2025 patients are included in this model.

## Data Availability

The University of Arizona clinical quality improvement database is not publicly accessible.

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
