# Peer review of "A Target for Increased Mortality Risk in Critically Ill Patients: The Concept of Perpetuity"

_jcm, 2021, doi:10.3390/jcm10173971_

Round 1

Reviewer 1 Report

I think this manuscript is a well-designed study, especially methodology is interesting. 

Major comments

I think this study included many kinds of patients that required intubation, such as stroke, right?

If possible, reasons for tracheal intubation should be disclosed. Please discuss if a mixed population contributed to your findings.

Minor comments

Introduction

OK

Methods

Why you chose 8mL/IBW?  Please explain with adequate references.

The data of SpO2 was collected, but no mention of when the data was collected.

Please describe how authors chose covariates. 

No sample size calculation was described.

Results

Table: What is acuity score? Please describe.

Figure2: Partially invisible.  Please revise it.

Figure 3, 4: Are the graphs adjusted for other confounding factors? Please clear that up.

The 2 in O2 should be a subscript.

Discussion

OK

Author Response

I think this manuscript is a well-designed study, especially methodology is interesting. We thank the reviewer for their comments.

Major comments

I think this study included many kinds of patients that required intubation, such as stroke, right? Yes, all non-trauma adult patients. We chose mechanical ventilation as the illustrating concept for perpetuity as lung protective tidal volumes have been shown to be largely agnostic of disease state.

If possible, reasons for tracheal intubation should be disclosed. Please discuss if a mixed population contributed to your findings. This has been added to the demographics table.

Minor comments

Introduction

OK

Methods

Why you chose 8mL/IBW?  Please explain with adequate references. Thank you for the question. Lung protective ventilation is generally thought of as 6ml/kg IBW, however the studies almost uniformly use 6-8ml/kg, including the ARMA trial where the median was 7ml/kg. Guidelines also recommend 6-8ml/kg. Thus, we chose 8ml/kg over 6 do avoid error that may occur from comparing largely “lung protective” (<6) with largely “lung protective” (7-8ml/kg). Our analysis compares lung protective tidal volumes by all definitions to not lung protective by all definitions. In addition, we had done a sensitivity analysis in our models using 6ml/kg and found similar results.

The data of SpO2 was collected, but no mention of when the data was collected. The SpO2 data were part of the triage vital signs. Added notation on page 4, section 2.6.

Please describe how authors chose covariates. Covariates were chosen from available data points that would indicate severity of illness (such as NIPPV use) or a potential confounder (such as age). Notation added to the text.

No sample size calculation was described.

We conducted a simple power calculation in order to gain insight into the number of participants needed to detect various hazard ratios corresponding to a one-hour increase in perpetuity time.  Our approach follows van Belle (2008, chapter 6, pg. 130) in assuming survival times are exponentially-distributed.  We assume we have two groups of participants, one with one hour longer in perpetuity than the other.  Then, with 1000 participants per group (2000 participants in total) and a 5% significance level, we will have ≥ 80% power to detect a hazard ratio less than 0.89 or greater than 1.13.  As a) hazard ratios of 0.89 and 1.13 are fairly small, and b) we collected data on around 2000 participants, we are confident that the study is well-powered.

van Belle, G. Statistical Rules of Thumb. (John Wiley & Sons, Inc., 2008). doi:10.1002/9780470377963.

Results

Table: What is acuity score? Please describe. Thank you for the question. How we determined acuity is defined in the first section of the online appendix. It is essentially at unit of time defined as the time of admission to the time of intubation. We have added this notation to the table.

Figure2: Partially invisible.  Please revise it. Fixed

Figure 3, 4: Are the graphs adjusted for other confounding factors? Please clear that up. Yes, they are adjusted. Added notation in the figure title/legend.

The 2 in O2 should be a subscript. Fixed

Discussion

OK

Reviewer 2 Report

This is a very interesting read and it highlights a complex concept that most ARDS studies and ITU in-reach ED ventilation studies have touched upon, however thus far was not looked at specifically under this lense of 'perpetuity'. Well done to your team.

Fundamentally the idea is not novel, but I really like the way it is analysed and presented here. I say 'is not novel' because of the widely accepted nowadays 'time on suboptimal treatment' effect on outcomes including survival and / or mortality, for many acute presentations (i.e. MI, stroke, bowel ischaemia, septic shock, ARDS, VILI etc.) which is a complex situation with several covariates / covariables and external / internal effects and interactions. It is physiologically proven that any delay of optimal 'gold standard' care for any presentation leads to poor outcomes, and that aspect is not novel. What is novel is that we have to understand the data in order to make system-wide changes in an effort to improve outcomes. A single change in one variant (Vt) would not be expected to massively change outcomes, and especially for ARDS / VILI it would be slightly shortsighted, when there is abundance of evidence that a multifactorial approach improves outcomes, and there is benefit from looking at all ventilation parameters (Vt, PEEP, FiO2, Compliance, fluid balance) as a whole, rather than individually (I don't want to call it a 'bundle' but you get my point. Without any intention to simplify your analyses, I think what your results highlight here is that, once again, systemic failures affect outcomes. I agree that time-varying effects should always be an integral part of Cox survival analyses, especially when investigating effects in outcomes from treatments/interventions with a known time-sensitive cause-effect, such as injurious invasive ventilation, and we as academics don't do this enough. Moreover, I 'get' the parallelism with the antibiotic timing in sepsis, and it might help the untrained / unfamiliar reader to sort of understand what you mean, however it is not an appropriate parallelism and it might create confusion. For example, as you know the category of antimicrobials (time-dependent killing vs. concentration-dependent killing) and the PK/PD has been a question in studies looking at sepsis outcomes and timings, complicating the analyses and causing some controversy (not everyone improves because they received carbapenem within 1 hour, in the same way that not everyone improves because we started 6ml/kg Vt immediately). Knowing who will benefit from what is the ever-lasting question. Additionally, Vt itself doesn't have many sub-parameters like the antimicrobials do (either it is 'injurious' as per current understanding, or it isn't). Also, there is a known correlation with poorer outcomes and injurious ventilation even as a single time-point (at commencement), which complicates matters in such analyses, as not all patients will have the same progression to a 'stiffer lung' despite how long they stay on the 'wrong settings'. I think your points are strong by themselves and there is no need for comparisons / effort to find similarities.

Overall, this is a large retrospective analysis with extensive statistical work with the correct method and concept, however it doesn't come without biases, even though the authors have done additional sensitivity analyses in an effort to mitigate some of them.

Strong points:

  • Interesting concept
  • Dual-centre study
  • Large N
  • Appropriate design and statistics performed

'Weak' points:

  • Retrospective analyses
  • Somewhat extensive assumptions in data manipulation with shifting of timepoints (although somewhat unavoidable to complete the time-varying Cox with steps)
  • Very heterogenous population
  • 11% of N with missing data
  • Gap of knowledge of the effect of interventions prior to the starting point chosen (pre-'perpetuity'). Some patients' outcome likely could have been predetermined if another intervention took place
  • Assumptions of extubation times leads to unavoidable noise
  • Most likely varying practices between the 2 sites
  • Vt given gravity without looking at PEEP and a combination of Vt / PEEP
  • Was the 'ideal height' appropriately used to calculate the correct protective Vt? I can't see this in the manuscript (only 'weight' which is incorrect) 
  • I can't see any important co-variables included, such as nurse-to-patient ratio, during what shift they were treated (daytime vs. nighttime staff) 
  • Why are we still not treating everyone with non-injurious modes of ventilation everywhere?

Comments:

It would be interesting to see how a model that appropriately includes PEEP level would perform. If you have the data could you consider showing that?

Interesting concept, but it is rather expected that for patients with low/moderate oxygen saturation, there is already an increased HR of death as they are generally 'sicker' at intubation. The multiple statistical outputs stemming from the analyses are generally challenging to interpret and it would not be sensible to attempt any clinical correlations. Please steer away from those.

The significant multifactorial limitations of the analyses are slightly underplayed.

Overall I think the concept is very interesting and I personally share their point of view. Should the authors satisfactorily make the suggested slight changes, it would be an interesting read for the critical care community.

Kind regards.

Author Response

This is a very interesting read and it highlights a complex concept that most ARDS studies and ITU in-reach ED ventilation studies have touched upon, however thus far was not looked at specifically under this lense of 'perpetuity'. Well done to your team.

We thank the reviewer for the comments.

Fundamentally the idea is not novel, but I really like the way it is analysed and presented here. I say 'is not novel' because of the widely accepted nowadays 'time on suboptimal treatment' effect on outcomes including survival and / or mortality, for many acute presentations (i.e. MI, stroke, bowel ischaemia, septic shock, ARDS, VILI etc.) which is a complex situation with several covariates / covariables and external / internal effects and interactions. It is physiologically proven that any delay of optimal 'gold standard' care for any presentation leads to poor outcomes, and that aspect is not novel. What is novel is that we have to understand the data in order to make system-wide changes in an effort to improve outcomes. A single change in one variant (Vt) would not be expected to massively change outcomes, and especially for ARDS / VILI it would be slightly shortsighted, when there is abundance of evidence that a multifactorial approach improves outcomes, and there is benefit from looking at all ventilation parameters (Vt, PEEP, FiO2, Compliance, fluid balance) as a whole, rather than individually (I don't want to call it a 'bundle' but you get my point. Without any intention to simplify your analyses, I think what your results highlight here is that, once again, systemic failures affect outcomes. I agree that time-varying effects should always be an integral part of Cox survival analyses, especially when investigating effects in outcomes from treatments/interventions with a known time-sensitive cause-effect, such as injurious invasive ventilation, and we as academics don't do this enough. Moreover, I 'get' the parallelism with the antibiotic timing in sepsis, and it might help the untrained / unfamiliar reader to sort of understand what you mean, however it is not an appropriate parallelism and it might create confusion. For example, as you know the category of antimicrobials (time-dependent killing vs. concentration-dependent killing) and the PK/PD has been a question in studies looking at sepsis outcomes and timings, complicating the analyses and causing some controversy (not everyone improves because they received carbapenem within 1 hour, in the same way that not everyone improves because we started 6ml/kg Vt immediately). Knowing who will benefit from what is the ever-lasting question. Additionally, Vt itself doesn't have many sub-parameters like the antimicrobials do (either it is 'injurious' as per current understanding, or it isn't). Also, there is a known correlation with poorer outcomes and injurious ventilation even as a single time-point (at commencement), which complicates matters in such analyses, as not all patients will have the same progression to a 'stiffer lung' despite how long they stay on the 'wrong settings'. I think your points are strong by themselves and there is no need for comparisons / effort to find similarities.

Thank you for your thoughtful comments. As you point out mechanical ventilation is a much more complex topic than just tidal volumes. We made a point to elaborate on this in our discussion, which due to word count limitations has progressively been abbreviated. While you are correct that it is not just tidal volume, but tidal volume normalized to lung compliance that is the injurious variable, we felt confident in tidal volumes as the variable given the consistency of findings of injury/harm across disease states. We have adjusted the language in the discussion lightly. Unfortunately, PEEP, airway pressures, nor lung compliance are available in our data set.

Lastly, the points you raise about the antibiotic parallel are noted. The intent of the parallel is the neglect of the second dose despite obsession in initiatives over the first dose. We have edited the text to this effect to try and strike the balance.

Overall, this is a large retrospective analysis with extensive statistical work with the correct method and concept, however it doesn't come without biases, even though the authors have done additional sensitivity analyses in an effort to mitigate some of them.

Strong points:

  • Interesting concept
  • Dual-centre study
  • Large N
  • Appropriate design and statistics performed

'Weak' points:

  • Retrospective analyses
  • Somewhat extensive assumptions in data manipulation with shifting of timepoints (although somewhat unavoidable to complete the time-varying Cox with steps)
  • Very heterogenous population
  • 11% of N with missing data
  • Gap of knowledge of the effect of interventions prior to the starting point chosen (pre-'perpetuity'). Some patients' outcome likely could have been predetermined if another intervention took place
  • Assumptions of extubation times leads to unavoidable noise
  • Most likely varying practices between the 2 sites
  • Vt given gravity without looking at PEEP and a combination of Vt / PEEP
  • Was the 'ideal height' appropriately used to calculate the correct protective Vt? I can't see this in the manuscript (only 'weight' which is incorrect) Ideal body weight was calculated based on height.
  • I can't see any important co-variables included, such as nurse-to-patient ratio, during what shift they were treated (daytime vs. nighttime staff). Nurse-to-patient ratio is not available. Day/night shift is added to the demographics table.
  • Why are we still not treating everyone with non-injurious modes of ventilation everywhere? Great question.

Comments:

It would be interesting to see how a model that appropriately includes PEEP level would perform. If you have the data could you consider showing that? This is an excellent suggestion, but unfortunately PEEP was not in our dataset.

Interesting concept, but it is rather expected that for patients with low/moderate oxygen saturation, there is already an increased HR of death as they are generally 'sicker' at intubation. The multiple statistical outputs stemming from the analyses are generally challenging to interpret and it would not be sensible to attempt any clinical correlations. Please steer away from those. We have edited to avoid the clinical correlations while still explaining the results in a way the reader can understand.

The significant multifactorial limitations of the analyses are slightly underplayed. Thank you for your comment. We believe we are very transparent about the significant limitations but are open to specific suggestions for further clarity.

Overall I think the concept is very interesting and I personally share their point of view. Should the authors satisfactorily make the suggested slight changes, it would be an interesting read for the critical care community.

Kind regards. We thank the reviewer for their support and thoughtful critiques. We hope the edits are satisfactory.